# Recent Advances in Electrochemiluminescence Sensors for Pathogenic Bacteria Detection

**DOI:** 10.3390/mi10080532

**Published:** 2019-08-13

**Authors:** Jinjin Shen, Ting Zhou, Ru Huang

**Affiliations:** MOE Key Laboratory of Laser Life Science & Institute of Laser Life Science, College of Biophotonics, South China Normal University, Guangzhou 510631, China

**Keywords:** pathogenic bacteria detection, electrochemiluminescence biosensor, nanomaterials, signal amplification, paper-based bipolar electrode

## Abstract

Pathogenic bacterial contamination greatly threats human health and safety. Rapidly biosensing pathogens in the early stage of infection would be helpful to choose the correct drug treatment, prevent transmission of pathogens, as well as decrease mortality and economic losses. Traditional techniques, such as polymerase chain reaction and enzyme-linked immunosorbent assay, are accurate and effective, but are greatly limited because they are complex and time-consuming. Electrochemiluminescence (ECL) biosensors combine the advantages of both electrochemical and photoluminescence analysis and are suitable for high sensitivity and simple pathogenic bacteria detection. In this review, we summarize recent advances in ECL sensors for pathogenic bacteria detection and highlight the development of paper-based ECL platforms in point of care diagnosis of pathogens.

## 1. Introduction

Pathogenic bacteria, such as Staphylococcus aureus (S. aureus), Salmonella typhimurium (S. typhimurium), Escherichia coli (E. coli) O157:H7 and Listeria monocytogenes (L. monocytogenes), etc., are great threats to human health and safety [1,2]. Raw materials and any processed foods may be contaminated by pathogenic bacteria. The World Health Organization (WHO) reported that nearly one in ten people worldwide suffer from foodborne illness, wherein 420,000 are cureless. Rapid and accurate diagnostic technologies for timely identification of the causative pathogens are crucial to optimize treatment, reduce mortality, and manage and minimize the spread of pathogens. Still, detection of pathogenic bacteria faces great scientific challenges and practical problems [3,4,5].

In general, the concentration of a pathogen is extremely low in the early stages of infection. Therefore, highly sensitive and specific detection systems are required for early diagnostic and surveillance purposes [6,7]. Although there are many reliable methods of pathogen detection, the need for trained and skilled personnel and sophisticated diagnostic instruments limits their application. Typically, a culture-based method is the gold standard for the diagnosis of pathogens in hospitals, but it is often laborious and time-consuming (could be several days or weeks), and not suitable for culture-negative bacteria. Alternatively, culture-independent biosensors can be viewed as cutting-edge technology since they offer transcendental advantages for the detection of bacteria, such as enzyme-linked immunosorbent assay (ELISA) and polymerase chain reaction (PCR); both of them can be combined with fluorescence, electrochemical, surface enhanced Roman scattering (SERS) and colorimetric assays [8,9,10,11,12] to quantify the pathogen concentrations.

Electrochemiluminescence (ECL) sensing method possesses excellent characteristics including, fast response speed, simple operation processes and excellent sensitivity, and has been widely used in the detection of proteins [13,14,15,16,17,18,19,20,21,22,23,24], nucleic acids [25,26,27,28,29,30,31,32,33] and metal ions [34,35,36,37]. Compared to fluorescence analysis, ECL does not require an excitation light source, thereby avoiding auto-fluorescence or a scattered light background. Moreover, using potential to initiate and regulate the output signal endows ECL sensors with high reproducibility and accuracy. Commercial ECL systems have been developed to detect many clinically important analytes, such as alpha fetoprotein (AFP), carcino-embryonic antigen (CEA), calcitonin and ferritin, etc. In recent years, the paper-based ECL detection platform has attracted much attention due to its great potential for point of care diagnosis [38,39,40].

In this paper, we review recent efforts for pathogenic bacteria detection based on the ECL assay, including novel ECL signal probe, the progress of main recognition elements of biosensors (including antibodies, antibiotic, lectin and aptamers), advances of the nucleic acid-based pathogen detection method, as well as the development of paper-based ECL platforms.

## 2. Electrochemiluminescence (ECL)-Based Detection of Pathogen

ECL is a process of electrogenerated chemiluminescence that combines the advantages of both electrochemical and photoluminescence analysis [41,42]. The classic ECL technique uses three electrode systems, including a working electrode, a reference electrode and a counter electrode. The electrochemical reaction taking place on the surface of the working electrode causes a specific chemiluminescence reaction. In brief, an illuminating substance forms an excited state illuminator by a highly energetic electron transfer reaction on the surface of the electrode. Then, the unstable excited state turns into the ground state and emits photos, according to the way free radicals are produced in ECL. It divides the quenching pathway and the co-reaction pathway [41].

### 2.1. ECL Probe

The emitter is an important intermediate for converting electrical signals into optical signals. Common emitters include luminol, quantum dots and ruthenium complexes. The transition metal compound Tris-(2,2′-bipyridine)ruthenium(II) (Ru(bpy)_3_^2+^) is the most commonly used ECL signal tag. The Ru(bpy)_3_^2+^/tripropyl amine (TPrA) ECL reaction mechanism has also been studied extensively. The ECL reaction of ruthenium complexes and TPrA is complex, and it is currently agreed that the ECL mechanism is based on an oxidation-reduction reaction model [43].
Tripropylamine − e^−^ → Tripropylamine·(1)
Ru(bpy)_3_^2+^ − e^−^ → Ru(bpy)_3_^3+^(2)
Ru(bpy)_3_^3+^ + Tripropylamine·→ [Ru(bpy)_3_^2+^]∗ + products(3)
[Ru(bpy)_3_^2+^]∗ →Ru(bpy)_3_^2+^ + hν(4)

The ECL reaction mainly involves two oxidation reactions on the surface of the working electrode. Firstly, TPrA is oxidized to form a cation radical of TPrA (TPrA^+^), and then the cationized TPrA rapidly loses a proton to produce a reductive TPrA (Equation (1)). Meanwhile, Ru(bpy)_3_^2+^ is oxidized on the same electrode to form a strong oxidizing Ru(bpy)_3_^3+^ (Equation (2)). Subsequently, the TPrA^+^ reduces the Ru(bpy)_3_^3+^ to form an excited Ru(bpy)_3_^2+^ ([Ru(bpy)_3_^2+^]∗, Equation (3)). Then, the excited-state Ru(bpy)_3_^2+^ decays to turn into the ground-state and emits photons at about 620 nm (Equation (4)). The resulting Ru(bpy)_3_^2+^ can be reused in subsequent reaction cycles as long as the TPrA is sufficient for the reaction to continue.

Introducing smart nanomaterials can further improve the signal intensity of ECL probes. For example, our group has constructed tris-(2,2′-bipyridyl) ruthenium labeled cysteamine to modify gold nanoparticles for the amplification of an ECL signal and achieved a limit-of-detection (LOD) as low as 100 fM for DNA detection [28]. Further, a linear Ru(bpy)_3_^2+^-polymer with a controllable ECL signal-amplifying ratio has been developed and reached a remarkable LOD of 100 amol for the detection of the HBsAg gene of the Hepatitis B virus, carcino-embryonic antigen and 16S RNA of *Listeria monocytogenes* [32]. The detection limit is defined as the target concentration that produces a net signal (total signal minus background) equal to three times the standard deviation of a series of background repeats [44,45,46].

As early as 2002, quantum dots (QDs) have been used for ECL [47]. Compared to luminol and Ru(bpy)_3_^2+^, QDs possess good stability, flexible modifiability and size-dependent optical properties [48]. Many kinds of QDs, such as CdTe, CdSe, CdS and Mn doped Ag_2_S QDs, have been extensively introduced to ECL biosensors for the detection of proteins, nucleic acids, toxins, etc. [49,50,51,52]. Additionally, labeling different targets with multicolor QDs enables multiplex ECL assays [53]. Also, graphene quantum dot (GQD)/carbon dots (CDs) is an ideal nanomaterial with ECL performance due to its unique optical properties of high photostability, high electrical conductivity and tunable excitation, as well as a large surface area to immobilize biomolecules [54,55,56,57]. Especially, N heteroatoms doping GQDs (N-GQDs) can dramatically alter the electro-optical properties of GQDs. Chen et al. [58] used synthetic N-GQDs, which have abundant carboxy groups and exhibited strong ECL activity with K_2_S_2_O_8_ as a coreactant to detect *E. coli* O157:H7 and obtained a LOD of 8 CFU/mL. Liu et al. [59] reported a boron nitride (BN) QD-based biosensor with an ECL resonance energy transfer (ECL-RET)/surface plasmon resonance ECL (SPR-ECL) sensing mode. In this study, BN QDs and gold nanoparticles (Au NPs) were used as ECL signal tags and a quencher, respectively. The recognition of the well-designed hairpin probe to the target DNA enlarged the distance between BN QDs and Au NPs, thereby the SPC-ECL effect replaced the initial ECL-RET effect and the ECL signal from BN QDs displayed about six-fold enhancement. As a result, as low as 0.3 pmol/L of the Shiga toxin-producing *E. coli* (STEC) gene can be detected.

### 2.2. Direct Detection of Pathogens

Some existing ECL sensors for pathogenic bacteria detection are listed in Table 1, including their mechanisms, features and LOD. Based on the detected target, the detection methods can be divided into two categories: direct pathogen detection and nucleic acid-based pathogen detection. Direct detection of pathogens can be achieved by bacteria-mediated specific conjugation of a signal probe, or bacteria specifically triggered signal amplification. Antibodies, antibiotics, lectin and aptamers can serve as recognition elements of bacteria detection biosensors. More importantly, they can be used to isolate the target from the entire clinical sample volume from patients [60], such as urine, saliva, blood or serum, which often contain undesired components that could disturb the recognition elements and the target pathogen or inhibit the following nucleic acid amplification process. Combining enrichment and concentration methods with biosensors would significantly increase the detection sensitivity in a clinical application [61,62].

#### 2.2.1. Antibody Based ECL Sensors

Antibodies are large Y-shaped proteins and are often applied as recognition elements to capture bacteria. A well-known example is ELISA (enzyme-linked immunosorbent assay), in which the specific reaction between the target pathogen and the antibody connects the pathogen to the enzyme (horse radish peroxidase, HRP) and then produces a color reaction through the enzyme and the substrate [58,68,69,70]. When HRP is replaced with Ru complex and the capture antibody modified on a working electrode, the limits of detection (LODs) observed for ELISA and the ECL immunoassay for BoNT/A were 12 and 3 pg/mL, and for BoNT/B, they were 17 and 13 pg/mL, respectively. Thereby, a more sensitive ECL biosensor can be constructed. In addition, the specific interaction between the Fc region of immunoglobulin G (IgG) and *S. aureus* can be used to detect *S**. aureu* as shown in Figure 1I [71].

The introduction of nanoparticles greatly simplifies the operation process of ECL biosensors and endows the latter with superior advantages. Fe_3_O_4_ nanoparticles are most widely used in the ECL biosensor system to capture and separate the targeted bacteria from complex matrices under an applied magnetic field. Guo et al. [63] constructed a Faraday cage-type ECL immunosensor based on multi-functionalized graphene oxide (GO), which is not only an ideal carrier for (2,2′-bipyridine)(5-aminophenanthroline)ruthenium (Ru-NH_2_) and antibodies, but can be developed to extend the outer Helmholtz plane (OHP) to enhance the ECL signal (Figure 1II). Generally, an electrochemical reaction occurs at a surface that is several nanometers from the working electrode. The interface between the electrode and the solution is called the electrical double layer including the interior Helmholtz plane (IHP) and OHP. In the electrode reaction kinetics, the region of solvent molecules and ions that are specifically adsorbed on the surface of the electrode is called the IHP, and the solvated ions center locus region closest to the electrode surface is called the OHP. In fact, only a very small quantity of analytes linked to ruthenium molecules can be contacted with the OHP to produce an ECL signal. Therefore, by extending the effective OHP region, the intermolecular ECL reaction could be more efficient. Compared with directly modifying Ru-NH_2_ onto the antibody, the multi-functionalized GO based biosensor can dramatically improve the LOD of *Vibrio vulnificus* (VV) to 1 CFU/mL.

#### 2.2.2. Antibiotic- or Lectin-Based ECL Sensors

Another strategy for direct pathogen detection uses antibiotic or lectin, which serve as effective recognition elements due to their robust binding capacity to bacteria. For example, vancomycin (Van) is a broad-spectrum antibiotic that binds to peptidoglycan on the surface of Gram-positive bacteria through five hydrogen bonds (dissociation constant approximately 1–4 μM) and counteracts the synthesis of the bacterial cell wall. Similarly, concanavalin A (Con A) is a plant lectin that binds to glycoproteins on the surface of Gram-negative bacteria cells. Compared to antibodies, antibiotics and lectins have the advantages of being easy to synthesize, stable, low cost, as well as convenient for storage and transportation.

Zhu et al. reported an integrated bacterial enrichment and ECL biosensor for the detection of the bacteria gene [72]. They modified vancomycin onto the surface of amino-functionalized magnetic beads by dehydration condensation reaction of amino and carboxyl groups to enrich *L**. monocytogenes* from the liquid samples (Figure 2I). Then the enriched bacteria underwent in situ genomic isolation and PCR amplification. The amplification products were quantified by ECL analysis. The results showed the functionalized magnetic nanoparticles achieved 90% capture efficiency when the concentration of *L**. monocytogenes* was 100 CFU/mL. Considering that the loading efficiency of vancomycin on magnetic nanoparticles was still unsatisfactory, Yang et al. [64] constructed multivalent brush-like magnetic nanoprobes (Van-PEG-PLL-MNPs), which were constructed by modifying magnetic nanoparticles with poly-L-lysine (PLL) and PEG, then Van was conjugated to the carboxyl of the PEG. Compared to Van-MNPs, multivalent brush-like magnetic nanoprobes have higher enrichment efficiency (94%) for *Listeria monocytogenes* (Figure 2III).

An ECL biosensor based on Ru-Con A for *E. coli* detection was reported by Yang et al. [73], as shown in Figure 2II. The Ru-con A complex was anchored onto the surface of the electrode by carboxyl-functionalized single-wall carbon nanotubes. The ECL intensity was reduced when *E. coli* O157:H7 bound to the probes on the electrode surface. This ECL biosensor for *E. coli* detection can be completed within 70 min, and the LOD reached 127 CFU/mL.

#### 2.2.3. Aptamer-Based ECL Sensors

An aptamer is a unique nucleic acid sequence that possesses robust and specific affinity for a target (e.g., protein, small molecule, cell, bacteria) that was selected from the oligonucleotide library by systematic evolution of ligands by exponential enrichment (SELEX) technology; it has attracted much interest as recognized elements in pathogenic bacteria detection due to the advantages of its low-cost, chemically stability and high affinity [74].

Although antibody modified electrode and magnetic nanoparticles are often used in ECL detection of pathogens, aptamer-based ECL biosensors for pathogen detection are rare. Hao et al. [65] reported an ECL biosensor that linked a -NH_2_ modified *E. coli* aptamer onto the surface of the electrode by AgBr nanoparticles (NPs) anchored in 3D nitrogen doped graphene hydrogel (3DNGH) nanocomposites (Figure 3). Due to the steric hindrance, the capturing target pathogen can significantly reduce the ECL intensity. The ideal recognition element for ECL-based pathogen biosensors should have a strong affinity and selectivity to target pathogens. Aptamers can easily be functionalized with -SH, -NH_2_ or -COOH, thereby providing a convenient way to immobilize aptamers to electrodes. Up to now, a large number of aptamer-based fluorescence, SERS and colorimetric assays have been developed [12,75,76]. Also, a variety of aptamer-based ECL biosensors have been constructed and used to detect proteins [77,78], small molecules [79,80] and even cancer cells [81,82].

### 2.3. Nucleic Acid-Based Detection

Compared to bacteria, protein or bacterial toxin-based pathogen diagnosis, detection of bacterial DNA or RNA possesses transcendental advantages of sensitivity, specificity and flexibility, and holds a greater potential for early diagnosis of pathogen infection. More importantly, nucleic acid-based detection can be used in genetic typing [83], thereby identifying the subtype of pathogenic bacteria. Moreover, detecting bacteria RNA can differentiate living from dead cells. As mentioned above, many strategies utilize specific recognition elements to capture and enrich the target pathogen, and they finally analyze the bacteria by amplifying its feature gene. Nucleic acid-based tests often employ enzymatic amplification to improve sensitivity, and PCR is commonly used [84,85,86]. Relatively, isothermal amplification methods can amplify the target gene with high sensitivity and specificity under a constant temperature condition, such as hyperbranching rolling circle amplification (RCA) [87,88], loop-mediated isothermal amplification (LAMP) [83,89,90] and nucleic acid sequence-based amplification (NASBA) [91]; therefore, it is more suitable for point of care (POC) diagnosis.

Our group described a magnetic capture and hyperbranching rolling circle amplification (HRCA)-based ECL method for sensitively detecting the *hlyA* gene of *L. monocytogenes* (Figure 4) [66]. In this study, we used the efficient exponential amplification of hyperbranching rolling circle amplification (HRCA) combined with biotin labeled primer and a tris (bipyridine) ruthenium (TBR) labeled primer. Then, the resulting HRCA products were captured onto streptavidin-coated magnetic beads and analyzed by ECL platform based magnetic beads, thereby determining the presence of the targets. The LOD reached as low as 10 aM of synthetic *hlyA* gene targets and about 0.0002 ng/μL of genomic DNA from *L. monocytogenes.* Recently, Liu et al. [92] proposed an ECL biosensor for detecting *Staphylococcus aureus* (*S. aureus*). First, asymmetrical PCR was utilized to amplify the conserved 16S rDNA of bacteria. Then the universal signal amplification (USA) probes that possess the same region complementary with Ru(bpy)_3_^2+^ labeled signal probe (Figure 4) and various regions that can hybridize with different regions of the PCR products, were used for signal amplification. As a result, the LOD reached as low as 100 fM. Subsequently, they further developed the USA probe into a branched DNA (bDNA) probe for more efficient signal amplification [93]. This strategy can detect 5 pM asymmetric PCR products, which is 1–2 orders in magnitude higher than that of the previous biosensor.

### 2.4. Paper-Based Bipolar Electrode for Pathogen Detection

Microfluidic paper-based analytical devices (μPADs) are portable, rapid and low-cost detection platforms. Whitesides’s group at Harvard University first proposed the μPADs [94]. They used photolithography to form hydrophilic and hydrophobic channels on patterned paper. Due to capillary action, the liquid samples migrate along the hydrophilic channel without the need of pumps. The system can simultaneously detect glucose and protein in 5 μL of urine based on the colorimetric method. Due to the advantages of simplicity, ability of capillary action and sample storage, many paper-based colorimetric analysis platforms have been built to detect nucleic acid, protein, etc. [95,96]. Benefiting from the advantages of malleability, foldability, as well as the ability to filter the sample, several paper-based platforms integrate purification of sample, nucleic acid amplification and detection [97,98,99]. In addition, paper detection platforms are potable, degradable and disposable; thus, they are friendly to the environment and very suitable for point of care diagnosis [100].

The three-electrode electrochemical devices can be used for ECL reactions. In the reaction cell, ECL probes can be enriched onto the surface of the working electrode by specific recognition of the capture probe anchored target to generate an ECL signal [101,102]. The emitted photons can be recorded by optical detectors, such as photomultiplier tubes and a camera. This system is commonly used for the detection of biological samples, such as tumor markers. ECL instruments manufactured by Roche have also been widely used in clinical testing [103,104,105]. Yet, the classic ECL system has some shortcomings. First, the tested samples require a large volume to achieve the purpose of micro-detection. Second, the magnetic beads used to concentrate the target onto the surface of the electrode inhibit partial ECL signal.

The paper-based ECL device enables analysis requiring only microliters of the sample [106,107]. Through electrodeposition and in situ growth methods, metal, such as platinum, gold and silver, can be used to form electrodes on the surface of the paper. Carbon ink also can be used to make the working electrode on paper by screen printing [108]. Wax-screen printing is often used to form the hydrophilic channels on filter paper. A 3D paper-based ECL device was reported by Yu’s group [109]. Figure 5I shows the carbon-working electrodes printed on cellulose paper by screen printing. Eight carbon working electrodes can diagnose four tumor markers in real clinical serum samples. The ECL reaction was triggered by 0.5 to 1.1 V at room temperature. This device may further development for point of care testing. In addition, a folding paper-based ECL devices was also constructed by Yan’s group [110] (Figure 5II). A paper-based DNA sensor was constructed for DNA detection with a LOD as low as 8.5 × 10^−^^18^ M. Disposable paper-based chips reduce the contamination of the reference and counter electrodes. Therefore, these paper-based ECL devices provide a new platform for highly sensitive detection of biological samples.

A bipolar electrode (BPE) is an electronic conductor located between the anode and the cathode and immersed in electrolyte solution (Figure 5III) [111]. One end of the BPE near the anode of the driving electrode is the cathode, and the other end near the cathodic of the drive electrode is the anode. When a direct current (DC) voltage is applied to the driving electrode, a potential difference is generated across the BPE, and oxidation and reduction reactions occur at each end of the BPE, respectively. Thus, the BPE does not require direct electrical connection to activate the electrochemical reactions at its poles. Compared with the three-electrode ECL system, the BPE only needs a DC power supply, and is thus more suitable for point of care (POC) diagnostics [112].

Xu’s group [113] developed a multi-channel pBPE-ECL for multiplex detection of pathogenic DNA (Figure 6I). When a target gene specifically binds to the capture probe covalently modified onto the BPE, the single probes labeled by Pt nanoparticles are also captured by forming a “capture probe-target-detection probe” sandwich structure. The Pt nanoparticles can catalyze the reduction of dissolved O_2_, thereby accelerating the oxidation of Ru(bpy)_3_^2+^ and TPrA. In this device, each detection unit has six independent channels and a common negative electrode, which can make the detection signal more stable. However, this method still requires a complicated electrode modification process.

Barton’s group [114] reported another ruthenium complex, [Ru(phen)_2_dppz]^2+^ (phen = 1,10-phenanthroline; dppz = dipyridophenazine), which is quenched in aqueous solution and displays intense ECL signal when intercalating into double-stranded DNA. The development of the “light-switch” molecule enables label-free and washing-free biosensing, and thereby can improve the stability and simplicity of the ECL detection system.

In Figure 6II, our group [67] introduced the “light-switch” molecule [Ru(phen)_2_dppz]^2+^ into the paper-based BPE-ECL (pBPE-ECL) analysis system for sensitive detection of the *hlyA* virulence genes of *L**. monocytogenes*. In the study, [Ru(phen)_2_dppz]^2+^ intercalates double-stranded DNA amplicons produced by PCR and generates ECL signals. As a result, 10 copies/μL of *L**. monocytogenes* genomic DNA can be detected. LAMP also can be introduced into the [Ru(phen)_2_dppz]^2+^-pBPE-ECL system to simultaneously detect two antibiotic resistance genes, *mecA* (coding for methicillin resistance via penicillin binding protein 2a) and *ermC* (coding for macrolidelincosamide-streptogramin resistance via rRNA methylases), with a LOD of 100 copies [97].

## 3. Conclusions

In summary, there is still an urgent requirement to further exploit sensitive and specific biosensors for the detection of pathogenic bacteria. ECL detection technology holds great potential for pathogen diagnosis. Developing a highly efficient ECL probe, selecting reliable recognition elements, and utilizing suitable nanomaterials to serve as ECL signal tags or as carriers for signal molecules, as well as to modify electrodes to advance ECL efficiency, would be helpful in improving the performance of the ECL biosensor. Especially, paper-based ECL detection platforms enable sensitive and portable POC detection. Sample volume, impurities, the test results’ reproducibility and test cost also should be considered in the biosensors. However, detecting trace pathogens from a complex sample is still a great challenge. Thus, further development of ECL biosensors will be beneficial to clinic diagnostics, food safety and environmental monitoring.

## Figures and Tables

**Figure 1 micromachines-10-00532-f001:**
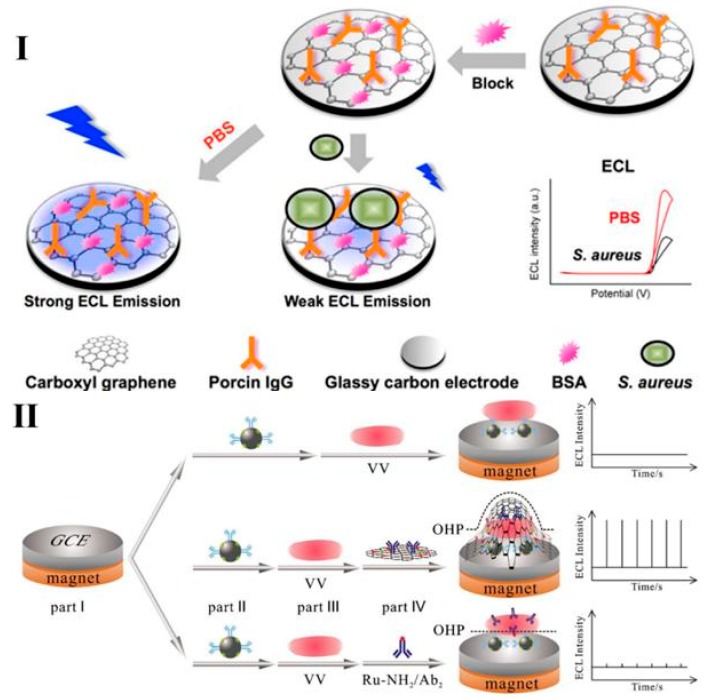
(**I**) Schematic of a label-free ECL pathway for *Staphylococcus aureus* (*S. aureus*) detection based on specific recognition between IgG and *S. aureus* protein A (SPA). Reproduced with permission from [71], published by Elsevier B.V, 2016. (**II**) Schematic of Faraday cage-type ECL immunosensor for detection of *Vibrio vulnificus*. Reproduced with permission from [63], published by Springer-Verlag Berlin Heidelberg, 2016.

**Figure 2 micromachines-10-00532-f002:**
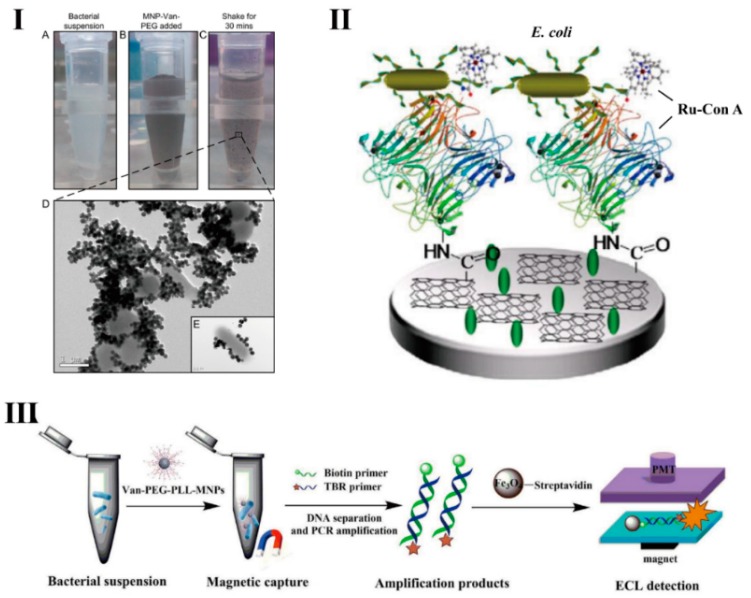
(**I**) Electron microscopic results of vancomycin-modified magnetic beads captured by *L**. monocytogenes*. Reproduced with permission from [72], published by American Chemical Society, 2015. (**II**) Schematic of ECL biosensor based on Ru-con A detector for *E. coli* detection. Reproduced with permission from [73], published by Elsevier B.V, 2012. (**III**) Multivalent brush-like magnetic nanoprobes (Van-PEG-PLL-MNPs) separation of *L**. monocytogenes* from bacteria suspension and DNA PCR amplification for ECL detection. Reproduced with permission from [64], published by Elsevier B.V, 2017.

**Figure 3 micromachines-10-00532-f003:**
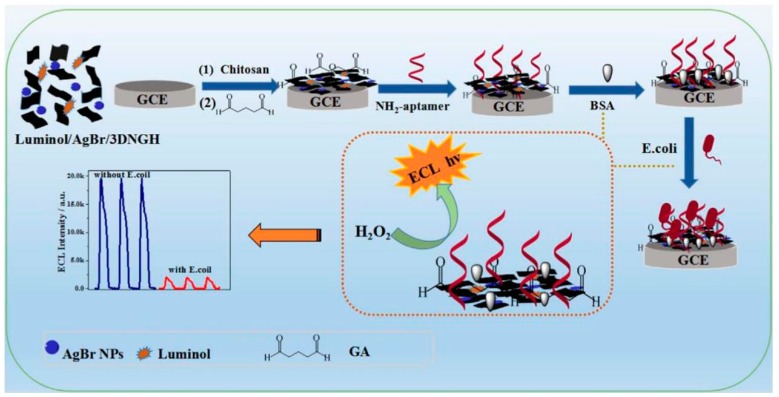
Schematic of ECL aptasensors based on AgBr nanoparticles (NPs) anchored 3D nitrogen doped graphene hydrogel (3DNGH) nanocomposites for *E. coli* detection. Reproduced with permission from [65], published by Elsevier B.V, 2017.

**Figure 4 micromachines-10-00532-f004:**
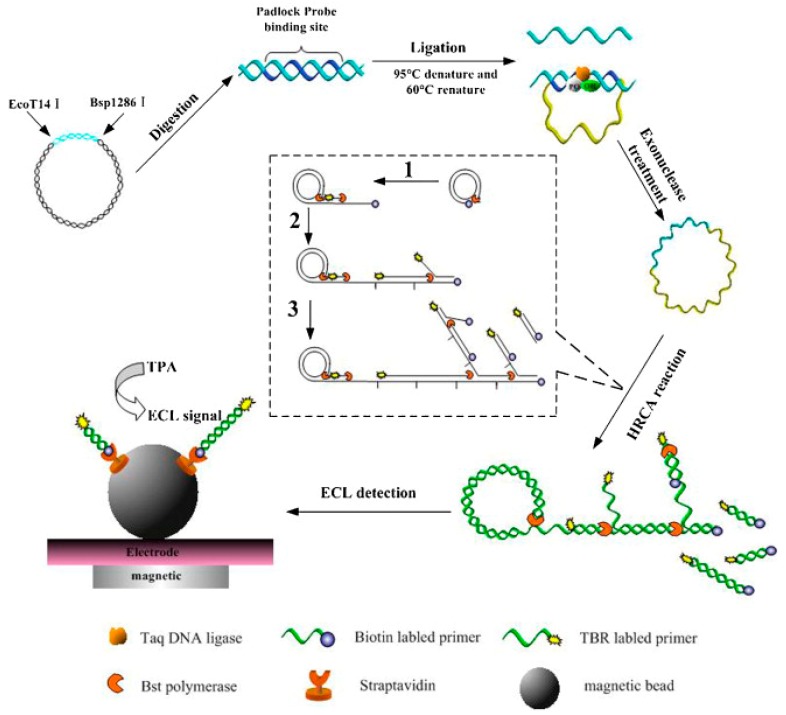
Schematic of isothermal ECL detection of *hly**A* gene of *L. monocytogenes*. Reproduced with permission from [66], published by Elsevier B.V, 2011.

**Figure 5 micromachines-10-00532-f005:**
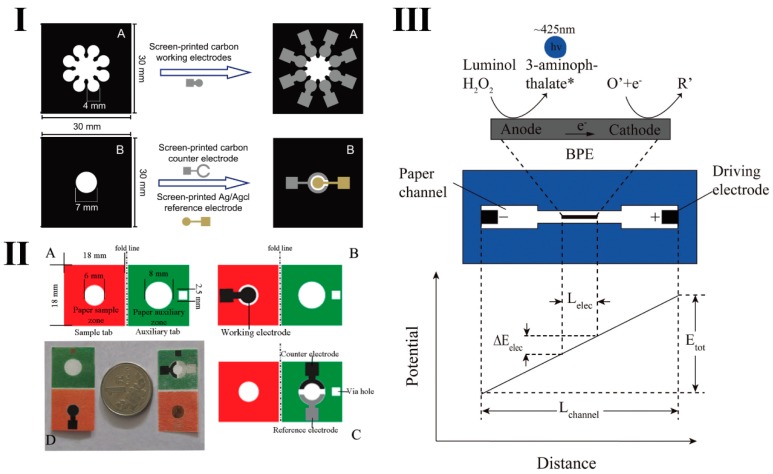
(**I**) Schematic of the process of making a 3D paper-based ECL device. Reproduced with permission from [109], published by Elsevier Ltd., 2012. (**II**) Schematic of folding paper-based three electrodes system. Reproduced with permission from [110], published by Elsevier B.V, 2014. (**III**) Schematic of ECL reaction of paper-based bipolar electrode. Reproduced with permission from [111], published by Elsevier B.V, 2016.

**Figure 6 micromachines-10-00532-f006:**
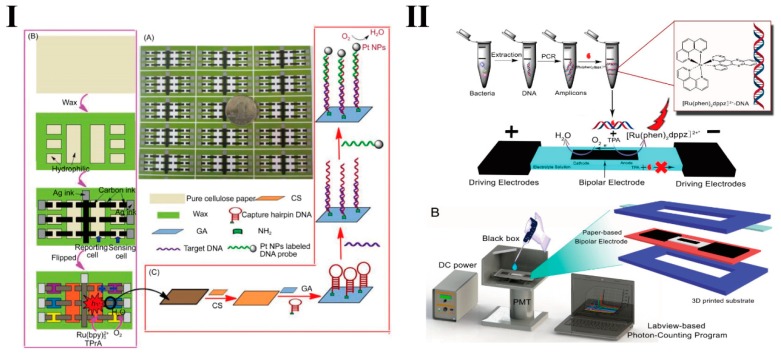
(**I**) Multi-channel paper-based bipolar electrode based on Ru(bpy)_3_^2+^ and TPrA reaction for pathogenic DNA detection. Reproduced with permission from [113], published by Science China Press and Springer-Verlag Berlin Heidelberg, 2015. (**II**) Schematic of paper-based bipolar electrode electrochemiluminescence (pBPE-ECL) analysis system for sensitive detection of pathogenic bacteria. Reproduced with permission from [67], published by American Chemical Society, 2016.

**Table 1 micromachines-10-00532-t001:** Existing ECL methods for pathogens detection.

Methods	Mechanism	Features	Limits of Detection	Detection Ranges	References
Antibody-based ECL sensors	Antibody based specific recognition of target and ECL signal probes.	A bacterium can connect multiple signal probes; Require specific antibodies to recognize the bacterial.	1 CFU/mL	1~4 × 10^8^ CFU/mL	[63]
Antibiotic- or lectin-based ECL sensors	Antibiotic/lectin-modified magnetic beads capture and enrich the target pathogen, thereby attaching the ECL probe.	Antibiotic/lectin is easy to synthesize, highly stable, low-cost, as well as convenient to store and transport; Poor specificity.	10 CFU/mL	10~10^4^ CFU/mL	[64]
Aptamer-based ECL sensors	The modified aptamer specifically captures the target pathogen onto the surface of the electrode.	Aptamer is easy to synthesize and modify; Selecting new and suitable aptamers is complicated.	0.17 CFU/mL	0.5 to 500 CFU/mL	[65]
Isothermal amplification-based detection	Isothermally amplify the characteristic gene of the target pathogen, then link the signal molecules/probes to the amplification products.	Employ enzymatic amplification to improve sensitivity; Require extracting the nucleic acid of pathogens.	0.2 pg/μL genomic DNA	0.2 pg/μL~2 ng/μL	[66]
pBPE for pathogen detection	When a direct current voltage applied to the driving electrode, a potential difference is generated across the BPE, and oxidation reaction and reduction reaction can occur at each end of the BPE, respectively.	Provide a potable, degradable and disposable detection platform of biological samples.	10 copies/μL	10~10^6^ copies/μL	[67]

ECL—electrochemiluminescence; pBPE—paper-based bipolar electrode.

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
