# Peer review of "Recent Advances in Electrochemiluminescence Sensors for Pathogenic Bacteria Detection"

_micromachines, 2019, doi:10.3390/mi10080532_

Round 1

Reviewer 1 Report

The paper is a review of the most advances methods to improve the electrochemiluminiscence sensors for pathogenic bacteria detection.

In general terms, I consider that the paper is well structure and writting, but I suggest to the authors that they would try to explain better the techinical parts of their work. I'm not an expert in the field of the paper, and I suppose that not all the readers of the journal are experts too (same as me). When the authors explain the expecific sensors that they analyze (nearly all the section 2), it was very complicate to me to follow the explanation without searching in internet some terms (Ru(bpy), for example).

I recomend that they must consider that the readers could not be experts, and they would try to explain better these important part.

Moreover, they will try to correct some "mistakes" with the way to cite the references in the text. For example, in line 83, thay cite the autor, but not apear the reference; same in line 85, 120... Check the line 62, because I supose that parenthesis are forgotten (they repeat ECL probes).

Reviewer 2 Report

Accept in present form

Reviewer 3 Report

1.       Overall, the writing is pretty good but there is a lack of transitions when going from one topic to another. This leaves out the relationship one has to the next and not connecting the ideas the authors are trying to present.

2.       Reactions 1-4 and figures 1 and 3 are not mentioned within the article. These should be used to expand the explanations of these concepts.

3.       I found that a lot of concepts discussed could have been expanded more to describe it’s importance and their relevance to ECL designs. For example, in line 80-81, why are GQD and CDs ideal for ECL? Line 108, changed to ECL is more sensitive but how much? What is the relevance of line 109-112? Line 151 how is it enriched? Line 207 transition from first sentence to the second is needed. The first sentence concept could be expanded. Line 245-247 each could be explained more.

4.        Some sections are missing references. For example, line 108-112. Lines 235-243.

5.       I felt the organization of section 2 could be improved. Line 117-133 is in the antibody-based ECL sensors but doesn’t seem to be about antibodies. This section also goes into great detail about IHP and OHP but that is pretty much the only section that explains the concept. I also think it would make sense for the problem to be explained followed by the explanation on how this is overcome and how it is efficient for ECL sensors. Line 228-232 is an oddly placed paragraph. Line 233-235 feels like it should be organized somewhere else. You shouldn’t be introducing a general topic on page 8.

6.       Double check references in bibliography, some have journals that use acronyms and some are full names.

Minor comments:

Line 7: change to “threats”

Line 8: change to “…choose the correct drug treatment…”

Line 11: change to “…but are limited…”

Line 12: change to “…biosensors combine…”

Line 13: suggested reword to “…suitable for high sensitivity and simple pathogenic bacteria detection.”

Line 15: change to “platforms”

Line 20: change to “bacteria”

Line 23: “accurate” was used twice in the same sentence. I would suggest replacing one.

Line 26: change to “challenges and practical problems.”

Line 27: suggest changing to “a pathogen”

Line 28: change to “…systems are required…”

Line 31-32: change to “…a culture-based method is the gold standard for the diagnosis of pathogens in hospitals, but…”

Line 35: change to “….(ELISA) and polymerase…”

Line 36: change to “…be combined…”

Line 41: change to “...does not require an excitation…”

Line 42: change to “…ECL sensors…”

Line 45: change to “In recent years,…”

Line 46: change to “…due to its great potential…”

Line 50: change to “…platforms.”

Line 52: change to “…combines…”

Line 54: change to “…by a highly…”

Line 55: change to “…and returns to…”

Lines 56-58: needs to reworded for better flow.

Line 60: change to “The/An emitter…”

Line 62: remove “electrochemical probe”

Line 63: change to “…non-aqueous and aqueous media.”

Line 65: change to “…Leland et al utilized tripropyl amine (TPrA) and ruthenium complex (Ru[bpy]32+)…”. It could also be “utilizes” just be consistent on the tense you use throughout the review.

Line 66-67: reword for clarity

Reaction 1: tripropylamine dot needs to be bold like in reaction 3

Line 70: change to “Our group has….”

Line 71: change to “….nanoparticles…”

Line 76: reword “…ECL effects of QDs…”

Line 76: extra period removed

Line 76: change to “Compared to luminol…”

Line 80: suggested change to “Additionally, labelling different targets with multicolor QDs enables multiplex ECL assays.”

Line 80: I suggest rewording to remove the “Also”

Line 84: change to “exhibited” and remove comma after coreactant

Line 85: space need after “(LOD)”

Line 85: what is the BN in the BN QDs?

Line 85: remove “ECL”

Line 86: change to “…surface plasmon resonance…”

Line 92 and 93: change to “pathogens”

Line 94-96: reword. Construct recognition element?

Line 96: Instead of “enrich target” would it be appropriate to say “isolate the target”

Line 98: “recognition” used twice in the same sentence. I suggest rewording so it is only used once.

Line 100: change to “…methods…”

Line 103: change to “…and are often applied…”

Line 104: change to “bacteria”

Line 104: change to “A well known example…”

Line 105: change to “…capture the target pathogen in the matrix…”

Line 104-107: needs to be reworded; it is described that two antibodies are used to capture and link the signal and an additional second antibody labelled is used an a signal probe.

Line 107-108: change to “When HRP is replaced with Ru complex and the capture antibody modified on a working electrode…”

Line 118: change to “…nanoparticles are…”

Line 119: change to “…separate the targeted bacteria…”. “targeted” is just a suggestion.

Line 122: change to “…but also can be developed to extend…”

Line 135: remove “as”

Line 148: suggested change to “…ECL biosensor…”

Line 157: change to “Yang et al”

Line 158: change to “…modified…”

Line 163-164: reword

Line 165: change “binding” to “bound to the probes”

Line 169: change to “An aptamer” and “…for a target…”

Line 172: change to “…attracting…”

Line 178: change to “…electrodes.”

Line 178: I think the way you currently have it worded the sentence starts better as “Up till now, a large…”

Line 181: change to “…nanoparticles are often used in ECL detection of pathogens…”

Line 182: change to “….biosensors for pathogen detection are rare.”

Line 183: change to “…anchored by/in…”

Line 184-187: reword

Line 194: “Entity”?

Line 197: change to “…used in…”

Line 198-199: change to “…living from dead cells.”

Line 199: change to “As mentioned above…”

Line 200: Change to “bacteria”

Line 201: change “typically example is” to “commonly”

Line 205: this is the only time POC is used and it not explained before.

Line 208: italics “Staphylococcus aureus”

Line 209-210: I believe this is the change that is needed for clarification “…with the complentary region to the Ru(bpy)32+ labelled…”

Line 210: change to “…hydridize with….”

Line 211: change to “…used for signal…”

Line 211: change to “As a results, the limit…”

Line 214: change to “S. aureus

Line 218: reword

Line 221: change to “pumps”

Line 222: reword “easy to modify”

Line 223: change to “…many paper-based colorimetric analysis platforms…”

Line 225: change to “….filter the sample, several paper platforms integrate…”

Line 226: remove “were constructed”

Line 226: change to “…platforms are…”

Line 227: change to “…friendly to the environment…”

Line 230: change to “…can be used to form electrodes…”

Line 230 and 231: add a space after the sentence

Line 233-234: change to “a working electrode, a reference electrode, and a counter electrode.

Line 235: change to “reactions”

Line 235: reword “…in the electrochemical reaction cell,”

Line 237: change to “a camera”

Line 240: change to “the classic ECL system”

Line 244: remove “several”. It doesn’t seem needed to me.

Line 245: Change to “A 3D…”

Line 245: change to “…group with a carbon…”

Line 246: remove “are”

Line 248: change to “…electrodes by the…”

Line 255: change to “A bipolar…”

Line 255: change to “…conductor located between the anode and the cathode…”

Line 257: change to “cathodic” and “anodic”

Line 259: change to “…oxidation and reduction reactions…”

Line 264: change to “….probe covalently modified onto the BPE, the single probes labelled by …”

Line 277: remove “but”

Line 284: replace “generated” maybe produced?

Line 285: what is LAMP?

Line 286: change to “detect”

Line 288: is it supposed to be 100 copies/uL?

Line 291: change to “biosensors for”

Line 295: change to “platforms enable”

Line 296: remove “of pathogen”

Line 296: change to “detecting”

Line 296-298: reword.

Line 298: change to “…ECL biosensors will be beneficial…”

Reviewer 4 Report

The topic is exciting and timely. However, severe grametical mistakes are present in the text, and the paper is not in the publishable state.  

Pathogenic bacterial contamination greatly threats human health and safety. How?

Define limit-of-detection and sensitivity. See for example,

Condens. Matter 2019, 4(2), 49; https://doi.org/10.3390/condmat4020049

&

Scientific Reports volume 6, Article number: 28077 (2016)

https://www.nature.com/articles/srep28077

Prepare a table showing detection limit and sensitivity

Define chemical symbols where necessary.

Page 8, Line 219: Define paper here.

Page 8, Line 239: Citation is a must.

Page 8, Line 245: Citation is a must here too.

Widespread grammatical errors and spelling mistakes are present in the text. The paper is not in the publishable state. It must be thoroughly revised.

Round 2

Reviewer 3 Report

Line 11: awkward wording "...are limited by complex and time-consuming."

Line 12: change to "biosensors"

Line 21: change to "bacteria". Bacterial is an adjective

Move sentence line 55-56 to after the next sentence. "....photoluminescence analysis. The classic ECL..."

Line 58: "...the working electrode causing..."

Line 61-63: not worded well

Move line 71 "on the surface of the working electrode" to end of line 70. Remove the rest of that sentence.

Line 75: should it be TPrA•?

Line 76: suggested wording "...decays to return to..."

Line 88: reword to "...QDs have been used for ECL." or something similar

Table 1: move to right before section 2.2.1

Line 110: I believe it should be worded as "...into two categories: direct and nucleic acid based pathogen detection."

Line 112-113: reword to "Antibodies, antibiotics, lectin and aptamers can serve as recognition elements of bacteria detection biosensors."

Figure 1: I would switch I and II so that they are discussed in numerical order within the test.

Line 157: change to "...serve as..."

Line 172: change to "...on magnetic nanoparticles..."

Line 186: change to "attracted"

Line 191: change to "...have a strong..."

line 212: change to "...commonly PCR."

Line 222: change to "The LOD..."

Line 226: suggested reword to "...possess the same region complementary..."

Line 228: change to "As a result,..."

Line 234: suggsted reword to "Whitesides's group of Harvard University first...."

Line 259: change to "...form electrodes..."

Line 263: remove "of"

Reviewer 4 Report

The authors have revised the manuscript and have addressed te comments provided to them to improve the quality of the paper. The paper should be ready for publishing following a careful proof reading of it. 

Author Response

We are very grateful for your endorsement to our work.